

# GAT TransPruning: progressive channel pruning strategy combining graph attention network and transformer

Yu-Chen Lin, Chia-Hung Wang and Yu-Cheng Lin

Department of Automatic Control Engineering, Feng Chia University, Taichung, Taiwan

## ABSTRACT

Recently, large-scale artificial intelligence models with billions of parameters have achieved good results in experiments, but their practical deployment on edge computing platforms is often subject to many constraints because of their resource requirements. These models require powerful computing platforms with a high memory capacity to store and process the numerous parameters and activations, which makes it challenging to deploy these large-scale models directly. Therefore, model compression techniques are crucial role in making these models more practical and accessible. In this article, a progressive channel pruning strategy combining graph attention network and transformer, namely GAT TransPruning, is proposed, which uses the graph attention networks (GAT) and the attention of transformer mechanism to determine the channel-to-channel relationship in large networks. This approach ensures that the network maintains its critical functional connections and optimizes the trade-off between model size and performance. In this study, VGG-16, VGG-19, ResNet-18, ResNet-34, and ResNet-50 are used as large-scale network models with the CIFAR-10 and CIFAR-100 datasets for verification and quantitative analysis of the proposed progressive channel pruning strategy. The experimental results reveal that the accuracy rate only drops by 6.58% when the channel pruning rate is 89% for VGG-19/CIFAR-100. In addition, the lightweight model inference speed is 9.10 times faster than that of the original large model. In comparison with the traditional channel pruning schemes, the proposed progressive channel pruning strategy based on the GAT and Transformer cannot only cut out the insignificant weight channels and effectively reduce the model size, but also ensure that the performance drop rate of its lightweight model is still the smallest even under high pruning ratio.

## INTRODUCTION

Deep learning and machine learning have made marked improvements in computer vision tasks in recent years. However, as the accuracy of the neural network increases, the computing performance and hardware required by the model also increase. Therefore, researchers have to design specific hardware to meet the computing needs of large networks, which demands substantial time and financial resources. Hence, deep neural network

Corresponding author
Yu-Chen Lin, yuchlin@fcu.edu.tw

compression has become a popular method for scaling the model size of the deep neural network. Many model compression methods exist, such as quantization, pruning, low-rank factorization, knowledge distillation, and reinforcement learning-based methods. Model compression methods involve downsizing and simplifying deep learning models while maintaining accuracy. These techniques are becoming increasingly important as the size and the demand for deep learning models in various fields continues to grow, and as the size of these models continues to increase.

However, the challenge with using large AI models today is that edge devices are resource-constrained. As neural network architectures become more complex, the number of parameters in the model can increase exponentially, which in turn requires more computational resources to train and deploy the model. This can make deploying these models in real-time applications difficult, where low latency and high throughput are important requirements. Edge devices, such as smartphones, Internet of Things (IoT) devices, and embedded systems, typically have limited memory and processing capabilities compared to traditional computing devices, such as desktop computers or servers. These limitations can pose significant challenges for deploying complex AI and machine learning models on edge devices, as these models often require large amounts of memory and processing power to execute. For example, running a large natural language processing model on a smartphone or IoT device may be impractical because of memory and processing limitations. Moreover, a bigger model means a higher inference time and energy consumption during inference. There are several reasons for compressing a deep learning model. For one, large models are computationally expensive and require more resources to train and run. This can be a significant challenge for applications that require real-time predictions or mobile or embedded devices with limited resources. Additionally, smaller models are often easier to deploy and use and can be more resistant to adversarial attacks.

For example, VGG-16 is a deep convolutional neural network architecture that has 16 layers, and it contains approximately 1.5 million nodes. The memory requirement of VGG-16 depends on the implementation, but it typically occupies around 528M runtime memory. The computational cost of VGG-16 is also high, with the classification of each image requiring up to 15 billion floating-point operations (FLOPs), as VGG-16 uses numerous convolutional filters and fully-connected layers, which require considerable computations to execute. Despite its high computational cost and memory requirements, VGG-16 has been widely used in a variety of computer vision applications, including image classification, object detection, and segmentation. However, because of its high computational cost and memory requirements, deploying VGG-16 on resource-constrained devices, such as smartphones or embedded systems, can be challenging. To overcome the challenges, researchers are exploring ways to optimize neural network architectures and reduce their computational requirements while maintaining high accuracy. Techniques such as model network pruning, quantization, and knowledge distillation can be used to reduce the size and complexity of neural networks while ensuring their performance. Additionally, hardware innovations, such as low-power neural network accelerators and efficient memory architectures are helping to reduce the energy and computational

requirements of these models. These techniques can reduce the size and complexity of the network, thus making it easier to deploy on resource-constrained devices, while still maintaining high accuracy.

In this study, a novel model compression method based on graph attention networks (GAT) and Transformer is proposed to determine the most important relationship in the channel, and an efficient pruned model. GAT TransPruning has the potential to significantly reduce the complexity and resource requirements of deep neural networks, while maintaining or even improving their performance, making them more practical for real-world applications. In conclusion, channel pruning is a powerful technique for compressing deep neural networks, making them more efficient, faster, and accessible for applications. Therefore, the main contributions of this study can be summarized as follows:

(1) *Combination of GAT and Transformer*: The proposed channel pruning method integrates the power of the GAT and Transformer to effectively identify and retain important channel relationships. This combination allows for a better preservation of model performance during pruning.

(2) *Learning-based approach*: Unlike traditional rule-based policies or human-engineered strategies, the proposed method leverages a learning-based approach to automate compression. This approach improves the effectiveness and innovation of model compression.

(3) *Comprehensive evaluation*: The effectiveness of the proposed method is thoroughly evaluated using popular network models, including VGG-16, VGG-19, ResNet-18, ResNet-34, and ResNet-50. The evaluation is conducted on the CIFAR-10 and CIFAR-100 datasets (*Krizhevsky & Hinton, 2009*), which allow for a comprehensive verification and quantitative analysis of the progressive channel pruning strategy. The experimental results demonstrate the efficacy of the proposed method. For example, applying a high channel pruning rate of 89% to VGG-19/CIFAR-100 decreases the accuracy rate by only 6.58%. Remarkably, the inference speed of the pruned model is 9.10 times faster than that of the original large model.

(4) *Suitable for edge computing*: The combination of the GAT and Transformer allows important channel relationships to be identified and preserved, thus ensuring that the lightweight pruned model maintains its performance. This makes it highly suitable for deployment on edge computing platforms with limited resources.

(5) *Automation and innovation*: The proposed progressive channel pruning strategy offers an efficient and innovative approach to model compression. It automates the design space sampling and improves the compression quality, unlike the traditional methods.

## RELATED WORKS

Model compression techniques have become increasingly popular in recent years as a method of accelerating and optimizing neural networks (*Dong et al., 2017*; *Luo, Wu & Lin, 2017*). These techniques involve reducing the size and complexity of a neural network without markedly sacrificing its accuracy or performance. Convolution implementations (*Bagherinezhad, Rastegari & Farhadi, 2017*; *Kim, Bae & Sunwoo, 2019*) and quantization

(*Gong et al., 2014*) can also accelerate the deep neural networks. Tensor factorization has also been utilized to decompose weights into lightweight pieces (*Masana et al., 2017*; *Yuan & Dong, 2021*; *Zhang & Ng, 2022*). As the field of artificial intelligence continues to grow and develop, the neural network models used for various tasks have become increasingly larger and more complex. While these models are often highly accurate and effective, they also pose significant challenges in terms of computational resources, energy consumption, and deployment. One of the primary drivers of the model size increase has been the pursuit of improved accuracy. Neural networks are typically trained using large datasets, and as the size of these datasets has grown, so too have the models needed to effectively learn from them. Additionally, the development of more complex architectures, such as convolutional neural networks and deep transformers, has allowed for the creation of more powerful models that are capable of handling increasingly complex tasks.

However, the larger a neural network model, the more difficult it becomes to train, optimize, and deploy. The computational power required to train and run these models is a major challenge. Training a large neural network on high-end hardware can take days or even weeks, and deploying these models in real-world scenarios can require considerable computational resources, which are often beyond the capabilities of many devices. To deal with these challenges, researchers have developed a range of techniques for reducing the size and complexity of neural network models. One popular approach is model compression, which involves reducing the size of a neural network model while preserving its accuracy. This can be achieved through techniques such as quantization, pruning, and knowledge distillation. Pruning involves removing unnecessary weights and connections from a neural network model, resulting in a smaller and more efficient model. Quantization involves reducing the precision of the weights and activations in a neural network, allowing for more efficient computation and storage. Knowledge distillation involves training a smaller "student" model to mimic the behavior of a larger "teacher" model, resulting in a smaller but still accurate model.

In addition to model compression, researchers have explored the use of compact architectures, which are designed to be inherently smaller and more efficient than traditional neural network models. These architectures often rely on specialized layers or structures that allow for more efficient computation and storage, while maintaining accuracy. Finally, joint optimization techniques have been developed to simultaneously optimize multiple aspects of a neural network model, such as accuracy, size, and energy consumption. These techniques often involve balancing trade-offs between these different factors to find the optimal model for a given application. As the neural network models used for artificial intelligence continue to grow and become more complex, the challenges associated with training, optimization, and deployment will only become more pronounced. However, techniques such as model compression, compact architectures, and joint optimization are being used by researchers to address these challenges and create more efficient and effective neural network models for a wide range of applications. Therefore, attention-based methods are used in this study to create an efficient pruning network.

## Model compression based on adaptive learning

Several studies have explored the use of reinforcement learning and genetic algorithms for searching efficient models in neural networks (*Brock et al., 2018*; *Miikkulainen et al., 2019*; *Szegedy et al., 2015*). AutoML (*He et al., 2018*) is a set of techniques that aim to automate the process of building, training, and optimizing machine learning models. It considerably improves the performance of deep neural networks by automating tasks that were traditionally done manually, such as hyperparameter tuning, model selection, and architecture design. A key benefits is that it can replace human effort in model compression and fully automate it. This is particularly useful in scenarios where human expertise is limited or expensive, or where the search space is too large for humans to explore manually. AutoML has been shown to achieve better results than humans in some cases, such as in the ImageNet classification challenge, where the systems have achieved state-of-the-art performance with markedly fewer parameters and computations than human-designed models. However, it is not a silver bullet and has its unique limitations. For example, AutoML may suffer from a high computational cost and may require large amounts of data and computing resources. DDPG (*Lillicrap et al., 2015*) receives the embedding state from a layer, t, which provides information about the current state of the layer, such as the number of input and output channels, the size of the feature maps, and other relevant statistics. The agent then outputs a sparsity ratio for the layer, which determines the percentage of channels that should be pruned. Once the sparsity ratio is determined, the layer is compressed using the selected ratio, and the network moves on to the next layer. This process is repeated for all layers in the network until all the layers are compressed. After the network is pruned, the accuracy of the pruned model is evaluated on a validation set, and a reward signal is computed based on the accuracy and the computational cost of the pruned network. Finally, as a function of accuracy and FLOP, the reward R is returned to the reinforcement learning agent. This reward signal is used to update the parameters of the reinforcement learning agent to improve its performance in selecting sparsity ratios that lead to high-performing and efficient networks.

Moreover, neural architecture search (NAS) (*Sekanina, 2021*) aims to automate the design of neural network architectures to achieve high accuracy while minimizing computational costs. One approach used in NAS is to search for transferable network blocks, which are reusable building blocks that can be combined in different ways to create a wide range of neural network architectures. The idea is to use an optimization algorithm, such as a genetic algorithm or a reinforcement learning algorithm, to search for the best neural network architecture for a given task by exploring a space of possible architectures. The search space can be defined in different ways, depending on the constraints and requirements of the problem. Transferable network blocks are advantageous, as they allow for a more efficient search process by reusing blocks that have been optimized for specific tasks and datasets. This can lead to faster convergence and better performance than searching for complete architectures from scratch, and its performance surpasses many human-designed deep neural network model architectures (*Chollet, 2017*; *He et al., 2016*; *Sandler et al., 2018*). In addition, N2N (*Ashok et al., 2017*) compresses a large trained teacher network into a small student network instead of defining a small network (student)

first. The model compression is then handed over to reinforcement learning, which is divided into two stages. The first stage determines which layers to delete, and the action is a binary variable to determine whether each layer should be retained or deleted, and a large, pre-trained "teacher" network is compressed into a smaller "student" network rather than designing a small network from scratch. Although model compression based on adaptive learning achieves better performance than that designed by a human, the process is still very time-consuming and energy-inefficient.

## Channel pruning

Pruning approaches can generally be categorized into unstructured and structured pruning. The main difference between them lies in the granularity of pruning weights. Unstructured pruning involves removing individual weights or neurons from the network without any constraints on their location or connectivity. This can be done by setting small weights to zero or removing entire neurons from the network. Unstructured pruning can be very effective in reducing the number of parameters and computations in a network; however, it can also lead to irregular sparsity patterns, which can be challenging to optimize and implement efficiently on hardware. In contrast, structured pruning involves removing entire filters, channels, or other structured units from the network while maintaining the connectivity and regularity of the remaining units. This can be done by setting entire rows or columns of weights to zero or removing entire filters or channels from the convolutional layers. Structured pruning can be more efficient in terms of reducing the number of computations and memory requirements, as the remaining units can be optimized and implemented more easily. In addition, structured pruning can lead to better generalization and transferability, as the remaining units are forced to learn more robust and transferable features. Both pruning methods have their unique advantages and disadvantages, and the choice of method depends on the specific requirements and constraints of the problem at hand. Unstructured pruning is generally more flexible and can achieve higher compression ratios; however, it can also be more challenging to optimize and implement. Structured pruning is more efficient and can lead to better generalization.

Some studies (*Zhang et al., 2018*; *Moon et al., 2019*) have focused on weight pruning, which is a common type of unstructured pruning and involves removing individual weights or neurons from the network without any constraints on their location or connectivity. Weight pruning is very effective in reducing the number of parameters and computations in a network, as it allows for fine-grained control over the sparsity level and can achieve very high compression ratios. However, weight pruning lead to irregular sparsity patterns, which is very challenging to optimize and implement efficiently on hardware. In addition, weight pruning results in increased sensitivity to the initial conditions and the optimization algorithm, as small changes in the weights have a large impact on the sparsity pattern and final performance of the network. The runtime acceleration is difficult to achieve because of irregular memory access (*Wen et al., 2016*), unless specialized hardware and libraries are used. Research has been conducted (*He, Zhang & Sun, 2017*; *Zhang et al., 2022*) to ensure structured pruning overcomes the problems mentioned above. Structured pruning has been proposed as a method of overcoming some of the challenges associated with

unstructured pruning, such as irregular sparsity patterns and difficulty in optimizing and implementing the pruned network. Structured pruning removes entire filters, channels, or other structured units from the network while maintaining the connectivity and regularity of the remaining units. This can be done by setting entire rows or columns of weights to zero or removing entire filters or channels from the convolutional layers. By removing entire filters or channels, structured pruning can produce a non-sparse compressed model, which can be optimized and implemented efficiently on hardware. Structured pruning is effective in reducing the size and computational requirements of deep neural networks while maintaining or even improving their performance. By removing the least important filters or channels, structured pruning can improve the generalization and transferability of the pruned network while reducing the risk of overfitting and instability and solving the problem of excessive time consumption associated with most model compression networks in training and inference.

Pruning is a general approach that can be applied to various deep learning tasks, such as image classification, object detection, and natural language processing. However, the search space of possible pruning configurations is very large, which makes it challenging to find an optimal configuration without relying on human expertise. Traditionally, pruning has been done manually, where experts would analyze the structure of the neural network and determine which neurons or connections to prune based on their knowledge and intuition. However, this process is time-consuming and often requires extensive trial and error. Automating pruning using these techniques can lead to marked improvements in performance and efficiency as compared to manual pruning, as the automated methods can explore a considerably larger search space and discover more optimal configurations. It can also save time and reduce the reliance on human expertise, making it more accessible to a wider range of researchers and practitioners. Recently, *Lin et al. (2017)* demonstrated a nascent approach for using reinforcement learning to perform pruning. The study used reinforcement learning to perform a sub-network selection during inference. However, the researchers did not really prune the network but selected a sub-network to draw the inference. *Yang et al. (2018)* and *Hooker et al. (2021)* pioneered the application of reinforcement learning in the context of pruning. However, their approach only provides rewards to the agent after an episode, resulting in sparse rewards and a lack of reinforcement at individual steps within each episode. This hinders the learning progress in the reinforcement agent, leading to a slowdown in its overall learning process. A review of the papers mentioned above revealed that the current model compression methods are not only very time-consuming to train but also energy-inefficient. Additionally, the most prominent obstacle towards structural pruning lies in the structural coupling, which not only forces different layers to be pruned simultaneously, but also expects all removed parameters to be consistently unimportant, thereby avoiding structural issues and significant performance degradation after pruning. To address this problem, *Fang et al. (2023)* proposed a general and fully automatic method, Dependency Graph (DepGraph), to explicitly model the dependency between layers and comprehensively group coupled parameters for pruning. In addition, existing works seldom extend the channel pruning methods to 3D point-based neural networks (PNNs). *Huang et al. (2023)* proposed CP3,

which is a channel pruning plugin for point-based network. CP3 is designed to leverage the characteristics of point clouds and PNNs in order to enable 2D channel pruning methods for PNNs. Specifically, it presents a coordinate-enhanced channel importance metric to reflect the correlation between dimensional information and individual channel features, and it recycles the discarded points in PNN's sampling process and reconsiders their potentially-exclusive information to enhance the robustness of channel pruning.

Moreover, vision transformer models have become prominent models for a range of tasks recently. These models usually suffer from intensive computational costs and heavy memory requirements. To alleviate this problem, *Yu & Xiang (2023)* proposed a novel explainable pruning framework dubbed X-Pruner, which is designed by considering the explainability of the pruning criterion. Specifically, to measure each prunable unit's contribution to predicting each target class, a novel explainability-aware mask is proposed and learned in an end-to-end manner. Then, to preserve the most informative units and learn the layer-wise pruning rate, X-Pruner adaptively search the layer-wise threshold that differentiates between unpruned and pruned units based on their explainabilityaware mask values. *Basha et al. (2024)* proposed a history based filter pruning method that utilizes network training history for filter pruning. Specifically, they prune the redundant filters by observing similar patterns in the filter's L1-norms over the training epochs. *Zheng et al. (2024)* identified how to evaluate the importance of filters as the key issue for the filter level pruning criteria to improve performance and proposed a new weight-based filter pruning method. The proposed method comprehensively considers the direct and indirect effects of filters, which can better reflect the filter importance, allowing the safe removal of unimportant filters.

In addition, most of the above-mentioned methods compress the model in a specific layer and rarely consider the relationship between layers. Determining the sparsity ratio for each layer in channel pruning can be a problem, as there is no one-size-fits-all solution that works well for all network architectures and datasets. One common approach is to use a validation set to evaluate the performance of the network under different sparsity ratios and choose the sparsity ratio that achieves the best trade-off between model size and performance. This approach can be time-consuming and computationally expensive, particularly for large networks and datasets. Therefore, channel pruning (*Chiliang et al., 2019*; *Chen et al., 2021*; *Liu et al., 2022*) reduces redundant channels from feature maps in convolutional neural networks, but how to determine the importance of each layer is the challenge. This study aims to design a novel channel attention mechanism that can determine and retain the important channel. 'Methods' describes how the pruning network works. This study adopted an attention mechanism to determine the interactive relationship between layers (*Tan & Le, 2019*), as the output of the previous layer would affect the output of the next layer. Therefore, this article proposes a high-efficiency structured channel pruning method and successfully identifies the interaction relationship between the channel weight, which targets import channels using the graph attention networks (*Veličković et al., 2017*) and Transformer (*Vaswani et al., 2017*) to retain the important channels and then remove a certain number of channels and the relevant filters to compress deep neural network models. This study uses the attention mechanism to find the elite channels in the model

and deletes the channels that are not important to the model to achieve the purpose of model compression, and finally, output the heat map as proof that the elite weights are retained.

## Dropout and federated learning

Two methods of model compression in deep learning are very similar to that in this thesis, which are dropout and federated learning. Dropout (*Srivastava et al., 2014*) is a regularization technique used in machine learning, particularly in deep learning models, to prevent overfitting and improve generalization performance. Overfitting occurs when a model performs well on the training data but fails to generalize to unseen data, leading to poor performance on new examples. In dropout, during training, a random subset of neurons or units is temporarily dropped out or set to zero with a certain probability. This means that these neurons do not contribute to the forward pass (output computation) or backward pass (gradient computation) during that particular training iteration. The process of dropping out neurons is done independently for each training example. The key idea behind dropout is to prevent the neural network from relying too much on any specific set of neurons. By randomly disabling neurons, dropout helps create a more robust and generalizable model that can handle variations and noise in the data. It can be thought of as an ensemble technique where different subsets of neurons work together to form multiple, diverse neural networks. During the inference or testing phase, dropout is usually turned off, and the entire network is used to make predictions. However, the final weights of the neurons are scaled down by the dropout probability to compensate for the increased number of active neurons during training. This scaling ensures that the expected output of neurons remains consistent between training and inference. Dropout has been widely adopted in various deep learning architectures, such as convolutional neural networks (CNNs) and recurrent neural networks (RNNs), and it has proven effective in reducing overfitting and improving the generalization capability of models.

Federated learning (*McMahan et al., 2017*) is a machine learning paradigm that enables the training of models across multiple devices or servers while keeping the data decentralized and distributed. It is designed to address privacy, security, and bandwidth constraints that arise in traditional centralized machine learning systems. In the traditional machine learning approach, data from various sources is collected and centralized on a single server or data center, where the model is trained. In certain applications, such as edge computing or IoT devices, real-time learning is required, and it may not be efficient to send data to a central server for model updates. Federated learning addresses these challenges by allowing the model to be trained directly on the devices or servers where the data is generated or stored, without centrally sharing the raw data. Each device performs local model training using its own data. During this phase, the device computes gradients based on its data and updates the global model. Instead of sending raw data to a central server, the devices only send the model updates (gradients) to a central coordinating server. Federated learning allows for distributed learning across multiple devices, making it feasible for edge devices and environments with limited connectivity. The above two methods are similar to that employed in this thesis. The method proposed in this thesis focuses on reducing the size

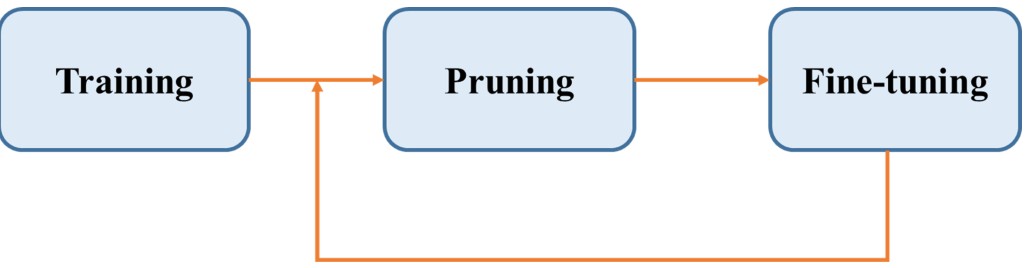

**Figure 1** **The general model pruning process.**

of the model, whereas dropout focuses on the overfitting problem. Moreover, federated learning mainly updates the model by integrating model updates from different devices. In this way, the model can continuously learn new data and knowledge from each device. Although both methods are suitable for running on embedded systems, the method utilized in this thesis aims to directly make the model lightweight rather than split the model into several small clients and then aggregate them, which occurs in federated laearning.

## METHODS

This article mainly proposes a novel model compression strategy with a channel attention module that we designed. The overall architecture of the proposed GAT TransPruning model is shown in Fig. 1. First, a large network needs to be trained, then pruned, and finally fine-tuned to restore the accuracy until a suitable small network is generated.

The proposed model compression strategy, which combines the GAT and Transformer, offers an effective solution to reduce the size and computational complexity of neural network models while preserving their accuracy. The key concept behind this strategy is to leverage the attention mechanism algorithm to identify and retain the most important channels for the neural network. During training, the network learns to assign importance scores to different channels based on their contribution to the overall model performance. These importance scores serve as the basis for selecting which channels to keep and which can be safely pruned. By removing unnecessary channels, the model achieves considerable reductions in size and computation without significantly compromising its performance. In the proposed channel pruning approach, the weights of the input convolutional layer are fed into the channel attention module designed for this purpose. The channel attention module consists of two main components: GAT and a Transformer network. The GAT is responsible for calculating the correlations between the weights as well as capturing the relationships and dependencies among them. The Transformer network then takes the correlation characteristics of the channel weights and computes the correlations between the channels in a different layer. To extract deeper features and enhance the representation power of the model, the output of the multi-head attention network is further processed through a feed-forward network. This additional transformation helps to capture more intricate patterns and relationships in the data. To prevent the network from becoming too deep and suffering from performance degradation, a residual module

is employed before inputting the features into the Transformer network and the feed-forward network. The multi-head attention network, feed-forward network, and residual module collectively contribute to extracting richer and more expressive features from the input data. This enhanced feature representation facilitates a better understanding and utilization of the channel relationships, enabling effective channel pruning. In summary, the proposed model compression strategy employs the attention mechanism, the GAT, and the Transformer to identify and retain the important channels in the neural network. Through the channel attention module, the correlations between the weights and channels are analyzed, and deeper features are extracted to enhance the representation power of the model. The combination of these components leads to significant model size reduction and computational efficiency improvements while maintaining satisfactory model performance. Further details are provided below.

## Network architecture

This study focuses on identifying and preserving important channels within a neural network by combining the GAT and Transformer models. The overall architecture of the proposed GAT TransPruning model is depicted in Fig. 2, and it consists of two main components: the large model and the channel attention module. The large model serves as the base network, and for the purpose of experimentation, popular CNN-based architectures, such as VGG and ResNet, are utilized. These large models usually contain a large number of channel weights; however, not all channels are meaningful, so large networks often suffer from an excessive memory capacity. The goal of this study is to prune unnecessary channels while maintaining network performance. To accomplish this, a progressive channel pruning strategy is devised. The channel attention module, represented by the light-yellow dashed box in Fig. 2, plays a crucial role in identifying the importance of channels across the entire network. This module is composed of two main components: the GAT network (blue block) and the Transformer network (pink block), both of which incorporate attention mechanisms. The GAT network operates on the channel weights of the large model by transforming them into channel weight nodes with ID information through the flatten operation. These channel weight nodes are then fed into the GAT network, which assigns attention scores to capture the relationships between the nodes. Higher scores indicate channels that are more significant and have to be retained by the network. However, the output of the GAT network only considers the channel relationships within a single CNN layer. To maximize the pruning ratio and account for inter-layer relationships, the output of the GAT network is further passed to the Transformer network in the pink block. The Transformer network employs a multi-attention mechanism to enable the cross-layer channel attention effects. This allows the model to capture deeper dependencies and relationships between channels across different layers. Finally, the results from the Transformer network are passed through a linear algorithm to calculate the loss. The loss function used in this study is typically the mean squared error (MSE) loss, which quantifies the discrepancy between the predicted and the target outputs. With the use of two attention mechanisms at different scales, the channel attention module can effectively identify channels that are truly important to the

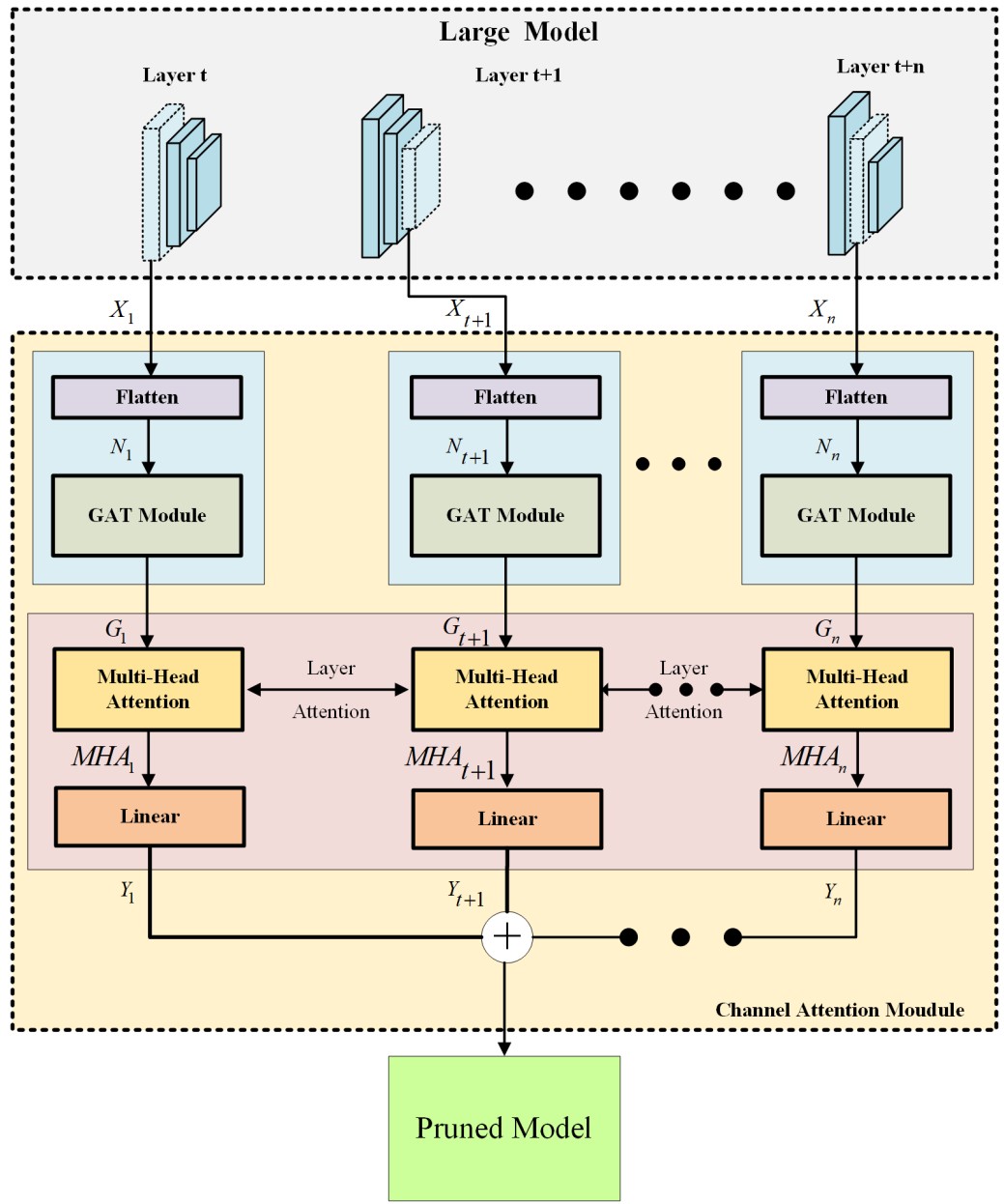

**Figure 2** The overall architecture of the proposed GAT TransPruning model.

network. The approach presented in this study combines the GAT and Transformer models to create a channel attention module within a larger neural network. By leveraging attention mechanisms at different scales, the module can identify and retain important channels while pruning unnecessary ones. This progressive channel pruning strategy allows for

more efficient and optimized models, reducing the model complexity without significant performance degradation.

Previously, the traditional pruning methods were mostly developed through manual design or pruning only for a single layer. Compared with the traditional pruning method, the pruning process utilized in this study is to first analyze the weight nodes of individual layers through GAT and then pay attention to the relationship of the channel weight nodes in different layers of the global network through the Transformer network. Thus, all of the layers are gradually pruned until they reach our preset standards. We called this layer-by-layer analysis and layer-by-layer pruning progressive pruning.

## Channel weight ID information

In the proposed progressive channel pruning strategy, it is crucial to accurately identify and prune unnecessary channels. Accordingly, a process is implemented to assign individual ID information to each channel before pruning. This process is illustrated in Fig. 3. The upper block in Fig. 3 represents the large model that is targeted for pruning. To establish unique identities for each channel weight, the channel weights are flattened. This flattening operation ensures that the pruning network can easily recognize and distinguish each channel weight as an individual channel weight node. Upon converting the channel weights of each layer in the large network into separate channel weight nodes, each channel weight is assigned a specific ID. Assigning individual IDs to the channel weights serves several purposes. First, it enables an efficient identification of channels that need to be pruned. With unique IDs associated with each channel, the pruning algorithm can quickly locate and target specific channels for removal. This accelerates the pruning process and streamlines the calculation of attention scores. Additionally, the assigned IDs are crucial in marking the attention scores for each channel. As the pruning network evaluates the importance of each channel, the attention scores can be associated with the corresponding channel weight nodes based on their unique IDs. This enables the attention scores to be accurately tracked and facilitates subsequent channel pruning decisions. By introducing individual ID information to the channel weights before pruning, this strategy enhances the precision and efficiency of channel pruning. The unique identities assigned to each channel enable faster identification of the channels to be pruned and streamline the calculation of attention scores. These ID-marked channel weight nodes are also crucial in accurately determining the importance of channels and facilitating the subsequent stages of the progressive channel pruning strategy.

As shown in Fig. 3, the subscript number of the symbol represents the layer in which the operation is performed. The formula for converting channel weights to channel weight nodes is as follows:

$$N = Flatten(X) \tag{1}$$

where $X$ represents the channel weight of the layer and $N$ is the result after channel weight $X$ be flattened. In the progressive channel pruning strategy, the channel weights of each layer in the large model are first flattened, resulting in a tensor $N$. This tensor $N$ represents the channel weights after flattening, which are further processed to facilitate the subsequent

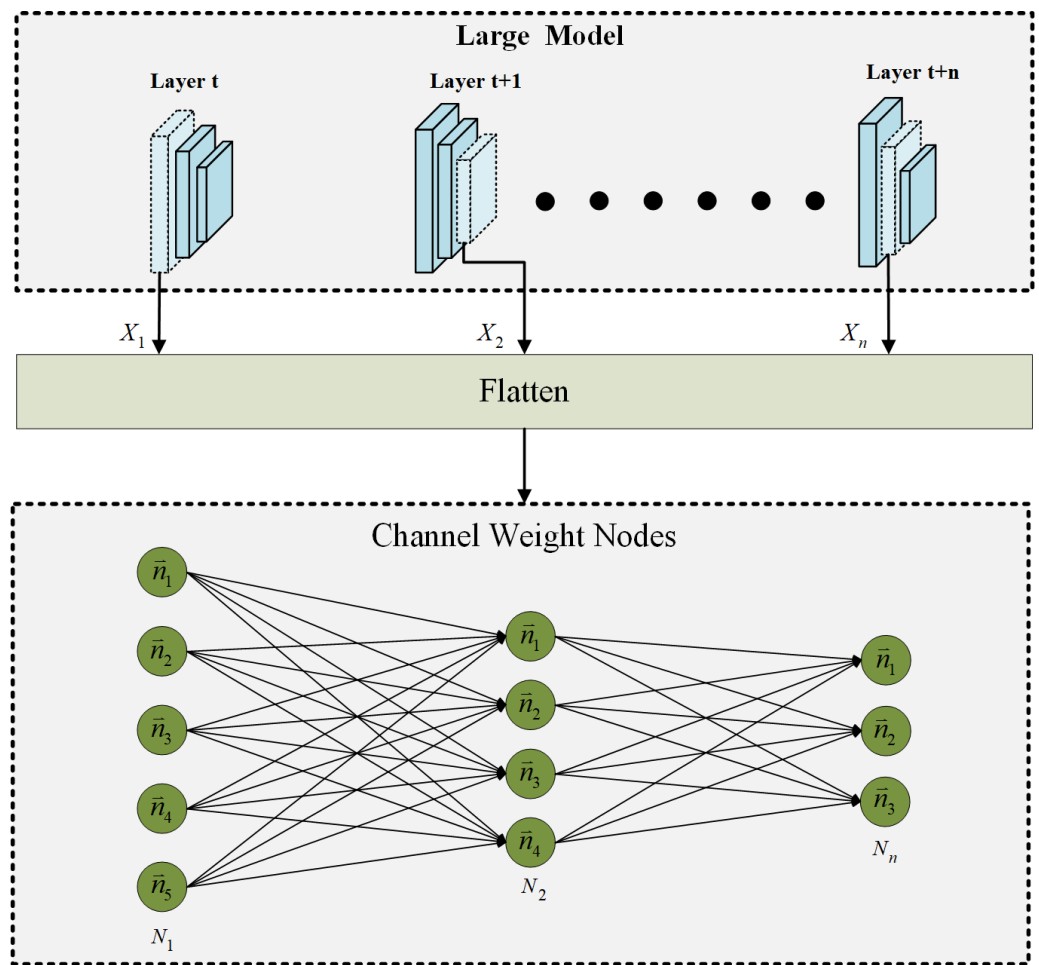

**Figure 3** The ID information of channel weight nodes *via* flatten.

attention calculation. To enable the attention calculation using the GAT, the flattened channel weights are converted into channel weight nodes with individual ID information. This ID-assigned representation allows for the accurate tracking and correlation analysis of the channel weight nodes. As depicted in Fig. 3, the correlations between the channel weight nodes are computed using the GAT module, which calculates the dependencies between the channel weight nodes based on their attention weights. The attention result $G$, which represents the attention weights computed by the GAT module, is determined using the following formula:

$$G = GAT(N). \tag{2}$$

Subsequently, the correlation features of the channel weights are inputted into the multi-head attention network. This network calculates the layerwise correlations among the channels. The multi-head attention network is followed by a linear transformation through the feed-forward network, which extracts deeper features from the output of the multi-head attention network. Finally, the pruning network generates a smaller model

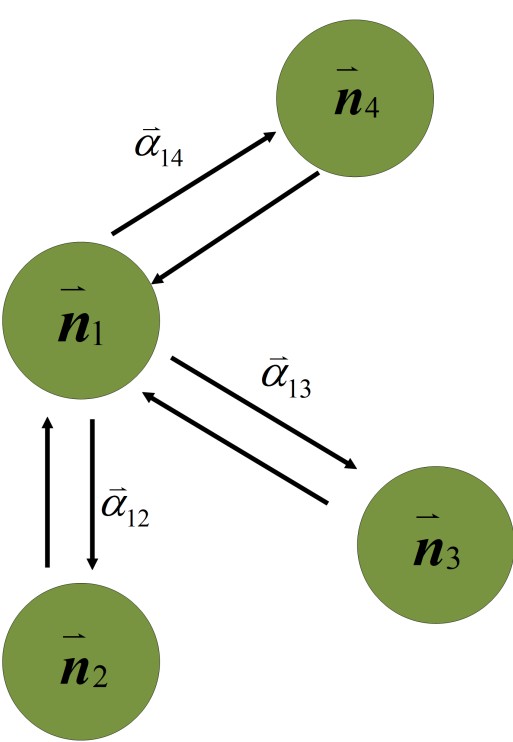

**Figure 4** The relationship between the channel weight nodes.

based on the output obtained from the Transformer. The transformation from the output of the Transformer to the small model can be expressed using the following formula:

$$MHA = FFN(Multihead(G)) \tag{3}$$

$$Y = Linear(MHA) \tag{4}$$

where *FFN* is the feed forward network, *MHA* is the multi-head attention process of calculating the layer correlation of the channel weight nodes, and *Linear* is linearly transformed. In these two formulas, the output of the Transformer, representing the result of the attention mechanism and the linear transformation, is used to generate the smaller model through the Transformer module. Through the aforementioned steps and formulas, the progressive channel pruning strategy leverages the GAT and Transformer modules to identify important channels and generate a smaller, pruned model while maintaining overall model performance. The attention mechanisms and correlation calculations enable accurate determination of channel importance, leading to effective channel pruning and model compression.

## GAT channel attention mechanism

To find meaningful channels for the network, this study uses the GAT to calculate the attention scores between nodes and eliminate unnecessary channels. GAT is a type of neural

network designed for learning representations of graph-structured data. For example, a social network can be represented as a graph, where nodes correspond to users, and edges represent friendships. The GAT is designed to learn representations of channel weight nodes in a large model that consider the structure of the network, as well as the features of the nodes and their neighbors. In GAT, each channel weight node is represented by a feature vector, which encodes information about the node itself. The attention score in GAT is computed using a learnable weight matrix, which is used to compute a similarity score between the features of each node and its neighbors. This similarity score is then used to compute an attention coefficient, which determines how much weight to assign to each neighbor when computing the representation of the node. Once the attention weights have been computed, the GAT computes a weighted sum of the features of the neighboring nodes using the attention coefficients as the weights. This weighted sum is then combined with the node's own feature vector, to produce a final representation of the node. This study uses the attention mechanism algorithm to find channels with better performance in large models and reserves them for channel pruning and explores the relationship between different channel weight nodes between layers. Therefore, the attention mechanism algorithm is mainly used to complete the progressive pruning task. As this study prunes the entire network and not just a single layer, channel pruning is performed from the first layer to the last layer, which is a progressive pruning method. As shown in the blue block in Fig. 2, after receiving the channel weight nodes with the ID information, the GAT calculates the attention weight between nodes for the channels in the network. Figure 4 shows a schematic representation of the GAT calculating the attention weight between nodesc. The GAT used in channel pruning improved the selection of important channels in a neural network. In particular, the GAT can compute importance scores for each channel weight node, which prune the least important channels. In the GAT, the network loads the channel weights of each convolution layers then flattens them into the input of the GAT module. The GAT input is a set of channel weight node features, $N = \vec{n}_1, \vec{n}_2, \ldots, \vec{n}_i, \vec{n}_i \in \mathbb{R}^F$, where $i$ is the number of channel weight nodes, and $F$ is the number of features in each channel weight nodes. The GAT output produces a new set of channel features, $N' = \vec{n}'_1, \vec{n}'_2, \ldots, \vec{n}'_i, \vec{n}'_i \in \mathbb{R}^{F'}$. At least one learnable linear transformation is required to obtain sufficient expressive power to transform the channel weight node features into higher-level features. In the first step, a shared linear transformation, parameterized by a channel weight matrix, $W \in \mathbb{R}^{F' \times F}$, is applied to the weight nodes of every channel. Then self-attention is performed on the channel weight nodes, which is a channel weight nodes shared mechanism, $a : \mathbb{R}^F \times \mathbb{R}^{F'} \to \mathbb{R}$ computes attention coefficients $e_{ij}$. The formula for importance of channel weight nodes is as follows:

$$e_{ij} = a(W\vec{n}_i, W\vec{n}_j) \tag{5}$$

where $i$ and $j$ is indicate that the importance of channel weight node $j$'s features to node $i$. Then, making the attention coefficients easily to compare across to different channel weight nodes, normalizing them across all choices of $j$ using the softmax function. The

formula is as follows:

$$\alpha_{ij} = soft\max(e_{ij}) = \frac{\exp(e_{ij})}{\sum_{k \in N_i} \exp(e_{ik})} \tag{6}$$

where $N_i$ is neighborhood of channel weight nodes of $i$. The normalized attention coefficients are used to compute a linear combination of the features corresponding to them to serve as the final output features for every node. In this article, GAT is a single-layer feedforward pruning network, so the coefficients computed by the channel weight nodes attention mechanism can be expanded as follows:

$$\alpha_{ij} = \frac{\exp(\text{LeakyReLU}(\vec{a}^T[W\vec{h}_i||W\vec{h}_j]))}{\sum_{k \in N_i} \exp(\text{LeakyReLU}(\vec{a}^T[W\vec{h}_i||W\vec{h}_j]))} \tag{7}$$

where $\vec{a}$ represents a weight vector, applying the LeakyReLU nonlinearity, $T$ represents transposition and $||$ is the concatenation operation. The schematic diagram of the relationship between the channel weight nodes is shown in Fig. 4. $\vec{\alpha}_{ij}$ is the attention score between the channel weights and the channel weights in the rest of the networks, and represents the influence between the channel weight nodes.

Therefore, GAT attention mechanism is used to determine the relationship between channel weights. Each channel is regarded as a node, and the network focuses on the most relevant channel neighbors of each node instead of treating all channel neighbors equally. In other words, this method can consider the degree to which channels influence each other, and even the relationship between channels among different layers. This is important because in a network, not all channels are useful, channel nodes have a large number of channel neighbors, and not all channel neighbors are equally important for the channel node representation. This mechanism is similar to the development of the human brain. The attention weights in GAT are computed using a learnable weight matrix, and this study uses this matrix for calculating the importance score between the weights of each node and its channel neighbors.

## Multi-head channel attention mechanism in layer level

This study determines the limitations of using attention weights solely from GAT as the criterion for channel pruning and acknowledges the need for a more comprehensive approach. While the GAT focuses on the attention weights within each individual layer, it fails to establish connections between layers. However, it is widely recognized that the CNN layer in a previous layer significantly affects the outcomes of the subsequent CNN layer. To address this issue, the article proposes a combination of the GAT and Transformer model to strengthen the interlayer connections and improve the effectiveness of channel pruning. The multi-head attention mechanism, a pivotal component of the Transformer model, plays a crucial role in achieving this objective. Originally designed for natural language processing tasks, the multi-head attention mechanism has demonstrated its versatility and found applications in various domains. In this study, the input sequence of channel weight nodes undergoes linear projections, resulting in multiple representations. Each representation is then fed into a separate attention head. Remarkably, each attention head

independently attends to different segments of the input sequence, enabling the model to learn distinct aspects of the relationships between channel weight nodes across different layers. This approach empowers the model to capture a diverse range of relationships and dependencies. The attention mechanism computes the attention weights for each channel weight node in the input sequence based on its relationships with the channel weight nodes from other layers. These attention weights determine the significance and relevance of each channel weight node with respect to the other layers. Leveraging multiple attention heads, the multi-head attention mechanism allows the model to simultaneously capture different types of relationships and focus on various aspects of the input sequence. With the incorporation of the multi-head attention mechanism, the Transformer model becomes proficient in effectively capturing dependencies and long-range relationships within the input sequence. It facilitates the parallel processing of different parts of the sequence, thereby enabling superior information aggregation and enhanced modeling of complex patterns. The integration of the multi-head attention mechanism within the channel pruning framework strengthens the inter-layer connections and enhances the model's expressive power. By enabling the Transformer model to capture diverse relationships and dependencies, this approach contributes to more accurate and efficient channel pruning. The findings of this study elucidate the importance of considering inter-layer connections and leveraging attention mechanisms to improve the performance of channel pruning techniques. This study goes beyond the conventional approach by considering the social interaction behavior between channel weight nodes in different layers. To achieve this, the attention mechanism of multi-head attention is employed to obtain importance predictions. The detailed process architecture is illustrated in Fig. 5. The input to the Transformer model is the output information of the channel weight nodes obtained from GAT. The Transformer model then processes the GAT outputs from different layers to compute the multi-head attention weights.

As depicted in Fig. 5, the input of the multi-head attention mechanism consists of the GAT results from three distinct layers. The Transformer model continuously selects and incorporates the GAT results from different layers to calculate the multi-head attention weights until the pruning network identifies the channels that need to be pruned. By thus leveraging the multi-head attention mechanism, this study aims to capture the intricate relationships and dependencies among the channel weight nodes across various layers. This approach allows for a more comprehensive analysis of the network dynamics and enables the identification of channels that may be pruned without compromising the model's performance. In summary, the proposed methodology utilizes the attention mechanism of multi-head attention to consider the social interaction behavior between channel weight nodes in different layers. By incorporating information from multiple layers, the model can effectively capture the intricate relationships and make informed decisions regarding channel pruning. The architectural design, as illustrated in Fig. 5, showcases the sequential process of obtaining multi-head attention weights from different layers until the target channels for pruning are identified. Multi-head attention is composed of multiple self-attention mechanisms, which are mainly used to extract the key information required

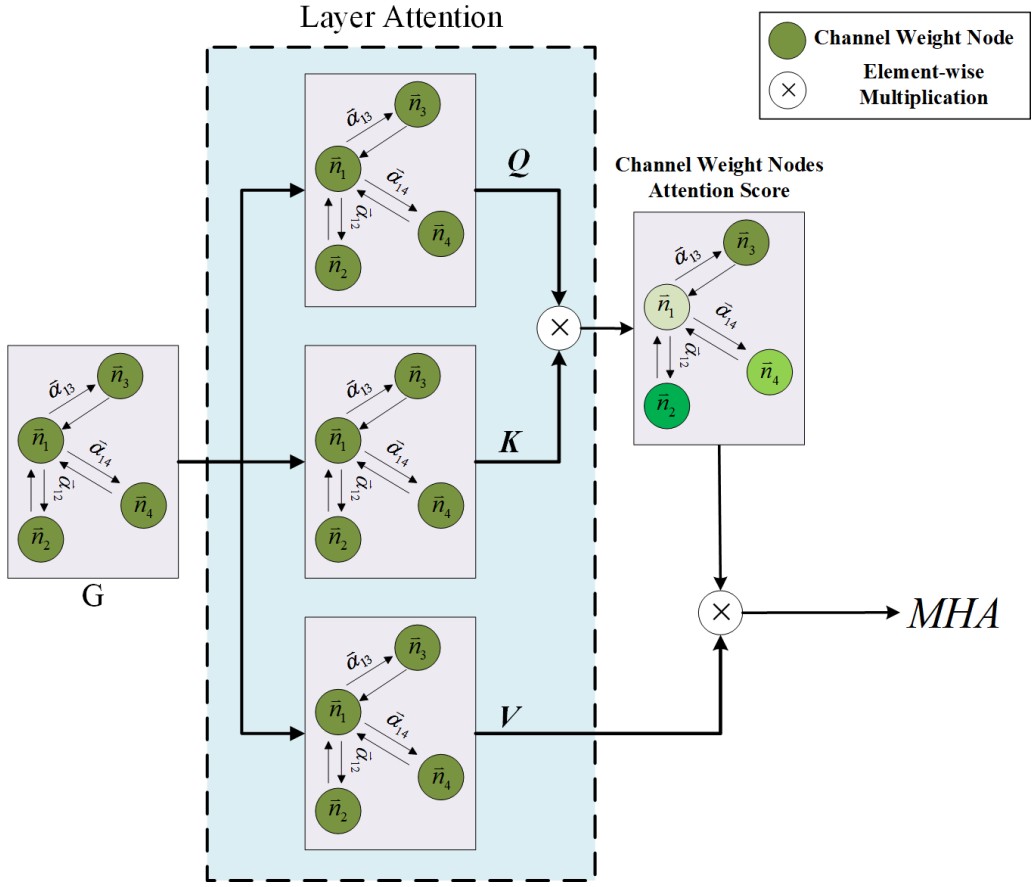

**Figure 5** The process of the channel multi-head attention.

in feature encoding. The formula is as follows:

$$\text{Attention}(Q, K, V) = \text{softmax}(\frac{QK^T}{\sqrt{d_k}})V \tag{8}$$

where $Q, K$ and $V$ refer to the channel weight vector from different layer, representing query vector $(Q)$, key vector $(K)$ and value vector $(V)$, which is used to score the importance of each channel. $d_k$ is the dimension of the hidden layer of $K$. By dividing by $\sqrt{d_k}$, the magnitude of the inner product can be prevented from increasing, which is conducive to model convergence. The multi-headed attention mechanism is defined as follows:

$$\text{MultiHead}(Q, K, V) = \text{Concat}(head_1, \ldots, head_h) \tag{9}$$

$$head_i = \text{Attention}(Q_i, K_i, V_i) \tag{10}$$

where head is one of the heads of the multi-head attention mechanism, which mainly divides the dimension of the input vector into equal parts according to the number of multi-heads The outputs of the multi-head attention vectors are then combined into a single vector through concatenation. Finally, the fully connected layer is used to map the combined multi-head attention vector to the output result. The multi-head attention

mechanism enables the model to learn and pay attention to the feature of its own channels and others channels through different angles of multiple heads and generate the channel feature required for pruning model predictions to enter independent self-attention layers for learning. The multi-head attention mechanism mainly utilizes different random initialization mapping matrices, that is, the fully connected layer. Finally, these heads are output and concatenated to perform a linear calculation. Each head focuses on different information, which is advantageous; some focus on local information while the others focus on global information. This ensures that the important channels are preserved. Through the aforementioned method, the pruning model can understand the input channel information from different layers and then obtain the attention score through the Softmax layer to retain the important channels and prune the less important channels to achieve model compression. Therefore, through the multi-head attention mechanism, the important channel feature codes of different layers are determined, and then, the network step is pruned by step to the highest compression ratio. This layer-by-layer pruning process is the progressive pruning strategy proposed in this article.

# EXPERIMENT RESULTS AND DISCUSSION

## Experimental settings and datasets

The experimental environment of this study comprised the National High Speed Network and Computing Center (TWCC). The graphics card GPU device used Nvidia Tesla V100 80GB and A100 from the laboratory; the operating system was Ubuntu 20.04; the CUDA version was 11.5; the cuDNN version was 8.3.0; and in the version of the deep learning kit Pytorch was 1.11.0. The number of training prunings in ResNet and VGG was 100 epochs, and its large model and pruning model were all operated in parallel. CIFAR-10 and CIFAR-100 dataset were used for the experimental process. CIFAR-10 comprises 10 distinct categories: airplane, automobile, bird, cat, deer, dog, frog, horse, ship, and truck. With a division of 50,000 training images and 10,000 testing images, it serves as a fundamental resource for assessing the capabilities of diverse deep learning models in image classification tasks. This dataset is important for its role as a standardized benchmark for evaluating the efficacy of different approaches within the realm of image classification. CIFAR-100, similar to CIFAR-10, is a dataset featuring a broader scope with 100 categories as opposed to 10. It comprises a collection of 60,000 $32 \times 32$ color images across these 100 classes, with each category containing 600 images. These classes are further organized into 20 superclasses, offering a structured hierarchy. Each image bears dual labeling: the specific fine-grained class it belongs to and the corresponding superclass. The dataset was divided into two subsets: 50,000 images for training and 10,000 for testing. CIFAR-100 serves as a formidable assessment tool for gauging the efficacy of deep learning models, especially when tackling intricate, fine-grained classification tasks. Its diversity and scale make it an ideal benchmark for testing the ability of models to distinguish between a multitude of subtle variations and intricate categories, thus contributing to the advancement of image classification techniques. The inclusion of superclasses introduces an additional layer of complexity, allowing researchers to delve into hierarchical classification strategies.

Table 1 **Comparison of pruning VGG-16 on CIFAR-10.** Comparison of pruning VGG-16 on CIFAR-10 for different pruning methods.

| Method | Accuarcy(%) | FLOPs | Parameters | Pruned(%) |
|---|---|---|---|---|
| VGG-16 | 93.96% | 313.73M | 14.98M | 0% |
| L1-norm (*Vaswani et al., 2017*) | 93.40% | 206M | 5.4M | 64% |
| GM (*Li et al., 2017*) | 93.58% | 201.1M | – | – |
| VFP (*He et al., 2019*) | 93.18% | 190M | 3.92M | 73.3% |
| SSS (*Zhao et al., 2019*) | 93.02% | 183.13M | 3.93M | 73.8% |
| FPEI (*Wang et al., 2021*) | 92.49% | 177.27M | 3.3M | 77.6% |
| GAL (*Lin et al., 2019*) | 90.73% | 171.89M | – | – |
| Ours-1 | 92.40% | 166.12M | 6.05M | 59.61% |
| HRank (*Lin et al., 2020*) | 91.23% | 73.7M | 1.78M | 88.1% |
| Ours-2 | 91.31% | 70.2M | 2.51M | 83.2% |
| Ours-3 | 90.21% | 45.82M | 1.61M | 89.3% |

## Model training

This study developed a strategy of channel pruning, which can achieve acceleration without the additional design of AI chips. It is inspired by the growth process of the human brain, which the human brain is known to prune and rewire its synaptic connections in response to changes in the environment. This process resembles the pruning process in a neural network. It learns through different inputs and finally deletes unnecessary neurons, multiplying into a high-performance small network. The neural network pruning method proposed in this article is a technique for reducing the model size and complexity of artificial neural networks without sacrificing network performance. This study conducted experiments on different CNNs models, such as ResNet18, ResNet34, ResNet50, VGG-16 and VGG-19. The experimental results revealed good performance from these different models.

## Experimental results and quantitative analysis

This article verified the proposed pruning strategy on VGG-16/CIFAR-10, and compared the results with those from other papers, as listed in Table 1. Ours-1, Ours-2, and Ours-3 represent our models under different pruning rates. Table 1 reveals that our method achieved exceptional results. It pruned most of the redundant channels and only decreased the model's accuracy marginally. As listed in Table 1, our smallest pruned model(Ours-3) removes 89.3% of the parameters while reducing accuracy by only 3.75%, and in our second pruned model(Ours-2), both the pruning rate and model accuracy are better than the results reported in other papers.

**Table 2  Comparison of pruning VGG-19 on CIFAR-100.** Comparison of pruning VGG-19 on CIFAR-100 for different pruning methods.

| Method | Baseline (%) | Accuracy After Pruned (%) | Δ Accuracy (%) | Speedup |
|---|---|---|---|---|
| OBD (*Wang et al., 2019*) | 73.74% | 60.70% | −12.64% | 5.73× |
| OBS (*Wang et al., 2019*) | 73.34% | 60.66% | −12.68% | 6.09× |
| EigenD (*Wang et al., 2019*) | 73.34% | 65.18% | −8.16% | 8.80× |
| Greg (*Wang et al., 2020*) | 74.02% | 67.55% | −6.67% | 8.84× |
| Ours-1 | 73.26% | 68.43% | −4.83% | 5.88× |
| Ours-2 | 73.26% | 66.68% | −6.58% | 9.1× |

**Table 3  Comparison of different pruning strategies on CIFAR-10 dataset.**

| Model | Accuracy (%) | Pruning ratio (%) |
|---|---|---|
| ResNet34 (Baseline) | 94.55% | 88% |
| L1-norm | 90.97% | 88% |
| L2-norm | 90.66% | 88% |
| Random | 90.37% | 88% |
| Ours | 91.29% | 88% |

This article also verified the proposed pruning strategy on VGG-19/CIFAR-100, and compared the results with other papers, as listed in Table 2. Ours-1 and Ours-2 represent our pruned models under different pruning rates. Table 2 reveals that the method had great result, which pruned most of the redundant channels and only decreased a few of the model's accuracy. As shown in Table 2, our second pruned model(Ours-2) accuracy only drops −6.58% and also gets the highest speedup ratio. The results of our pruned model are better than other papers.

In addition, this study also conducted experiments to further validate the effectiveness of the proposed method. Taking ResNet34/CIFAR-10 as an example, the article compared the performance of L1-norm, L2-norm, random pruning, and the proposed GAT TransPruning method. The results, as shown in Table 3, clearly demonstrated that the method proposed in this article achieved the highest accuracy under the same pruning conditions.

## Channels of interest analysis with heatmaps

To know exactly whether each pruning process pays attention to the same channels and keeps them in the next pruning epoch, this article pulled out the attention score of multi-head attention and made a heat map for deeper analysis. The heatmap shown in Fig. 6 is pruned on ResNet18. The Figs. 6A and 6B show the channel attention area that the pruning model focuses on for different pruning ratios. The areas and contours of attention are almost the same on the heatmaps of different pruning rates for the proposed method,

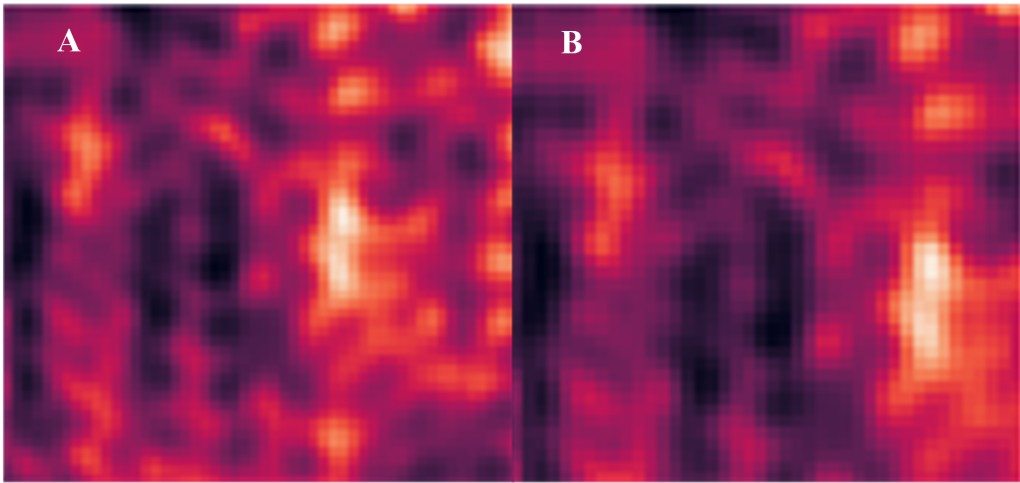

**Figure 6  Channel attention heatmap under different pruning ratios.** (A) Layer 0 at 48% pruning rate. (B) Layer 0 at 72% pruning rate.

as shown in the figure, which means that the model does repeatedly focuses on important channels. This proves that the pruning model does retain important channels rather than randomly removing unnecessary channel weights.

The channel attention heatmap has the same effect on ResNet34, ResNet50, VGG-16 and VGG-19 when visualizing the removed channel weights. The details of the channel attention heatmap are shown in Fig. 7. Figures 7A to 7H show the channel attention area that the pruning model focuses on for different pruning ratios for ResNet34, ResNet50, VGG-16, and VGG-19. The visualized result graph clearly shows that the important channel weights are determined at the beginning, and the important channel weights are indeed represented in subsequent pruning. The color depth represents the importance of the channel weights. The attention distribution in heatmap can be used to accurately determine whether each pruning process considers the same channels. Figures 7E and 7F display the channel attention regions of the pruned model at different pruning ratios. It is evident from the figures that the attention distribution in the heatmaps remains vertical across different pruning rates, effectively demonstrating that the pruned model indeed preserves important channels.

## Ablation experiments

In addition to the main experiments, this thesis also conducted ablation experiments to further validate the effectiveness of the proposed method. This article splits the pruned modules into different combinations to verify that the pruning strategy designed in this article is the best method. First, as listed in Table 4, the inter-module is disassembled to only use GAT, Transformer, and our designed method GAT TransPruning pruning strategy. Table 4 reveals that under the same pruning rate, the pruning strategy designed in this article obtains the best results, and the same pruning rate has the lowest accuracy drop. In addition, the accuracy of the three models, using only GAT and only Transformer

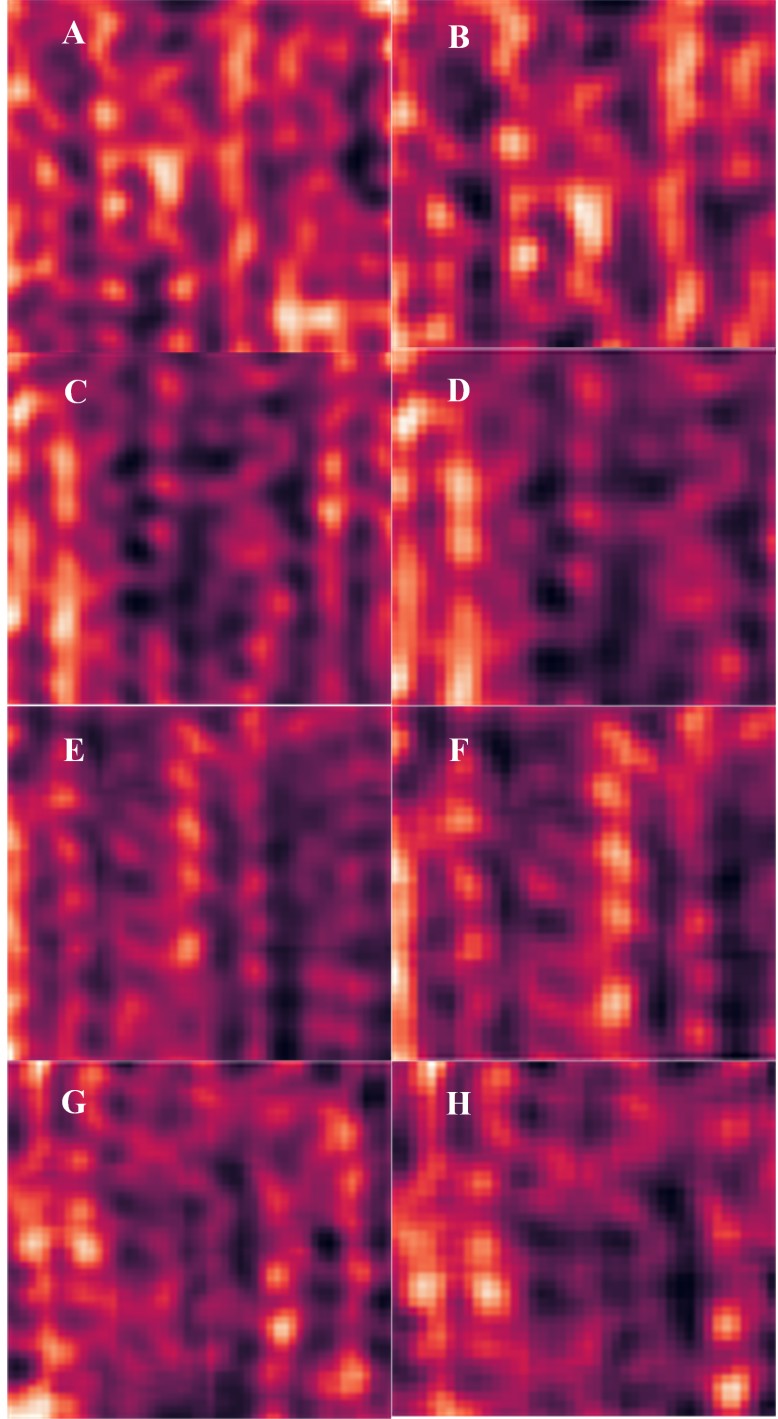

**Figure 7  Channel attention heatmap under different pruning ratios.** (A) ResNet34 layer 0 at 48% pruning rate. (B) ResNet34 layer 0 at 72% pruning rate. (C) ResNet50 layer 0 at 54% pruning rate. (D) ResNet50 layer 0 at 75% pruning rate. (E) VGG-16 layer 0 at 35.37% pruning rate. (F) VGG-16 layer 0 at 58.50% pruning rate. (G) VGG-19 layer 0 at 35.50% pruning rate. (H) VGG-19 layer 0 at 59% pruning rate.

**Table 4  Ablation experiment of GAT TransPruning strategy on CIFAR-100 dataset.**

| Model | Accuracy (%) | Pruning ratio (%) |
|---|---|---|
| ResNet18 (Baseline) | 92.3% | 99% |
| GAT Only | 90.40% | 99% |
| Transformer Only | 90.46% | 99% |
| Ours | 90.60% | 99% |

models decreased by 0.2% and 0.14% compared to the complete GAT TransPruning model, respectively, at the same pruning rate.

## CONCLUSIONS

Channel pruning is a popular model compression technique that aims to reduce the number of channels in a deep neural network, thereby reducing its computational complexity and memory requirements while maintaining or improving its accuracy. This technique has attracted significant attention in recent years because of its ability to significantly compress deep neural networks, making them faster and more efficient, which is essential for real-world applications, such as mobile devices and embedded systems. In this study, the powerful GAT attention mechanism was used for model pruning. GAT is a powerful technique used for channel pruning. It is a neural network that leverages the graph structure of data to perform feature aggregation and learning. It is highly effective in channel pruning with the use of channel attention mechanisms that improve the network's overall performance while reducing the number of channels. The contributions of this study are as follows: (1) When the users provided different pruning rate goals, the GAT TransPruning model found the best small model. (2) Improved computational efficiency: GAT TransPruning significantly reduced the number of parameters and computation costs of deep neural networks, making them more computationally efficient, particularly for deployment on edge devices with limited computational resources. (3) Improved memory efficiency: Pruned models had a smaller memory footprint, making them easier to store and transfer. (4) Better generalization performance: Pruning could improve the generalization performance of deep neural networks by reducing overfitting, particularly when the pruned models were fine-tuned. (5) Automated model compression: The use of neural architecture search techniques and deep reinforcement learning could automate the process of channel pruning, reducing the need for manual intervention and making the process more efficient. (6) Transferability: Pruned models could also be used as transferable network blocks for other models or tasks, making them useful for transfer learning. Overall, GAT TransPruning has the potential to significantly reduce the complexity and resource requirements of deep neural networks, while maintaining or even improving their performance, making them more practical for real-world applications. Channel pruning is a powerful technique for compressing deep neural networks, making them more efficient, faster, and more accessible for real-world applications. Future research should focus on developing more efficient and accurate pruning algorithms and investigating the use of channel pruning in larger and

more complex networks. Channel pruning is a promising area of research, and it is likely to play a significant role in the development of more efficient and powerful deep neural networks in the future.

### Funding
This work was supported by the National Science and Technology Council, Taiwan, R.O.C. (No. 112-2218-E-035-001). The funders had no role in study design, data collection and analysis, decision to publish, or preparation of the manuscript.

### Grant Disclosures
The following grant information was disclosed by the authors:
National Science and Technology Council, Taiwan, R.O.C.: 112-2218-E-035-001.

### Competing Interests
The authors declare there are no competing interests.

### Author Contributions
- Yu-Chen Lin conceived and designed the experiments, analyzed the data, prepared figures and/or tables, authored or reviewed drafts of the article, and approved the final draft.
- Chia-Hung Wang conceived and designed the experiments, performed the experiments, analyzed the data, performed the computation work, prepared figures and/or tables, and approved the final draft.
- Yu-Cheng Lin performed the experiments, analyzed the data, performed the computation work, authored or reviewed drafts of the article, and approved the final draft.

### Data Availability
The data is available at the CIFAR-10,CIFAR-100 from the Canadian Institute for Advanced Research (CIFAR), Canadian: https://www.cs.toronto.edu/~kriz/cifar.html.
Our code is available at GitHub and Zenodo:
– https://github.com/fcuace428/GAT-TransPruning.
– Fang, G. (2024). Torch-Pruning (v1.0.0). Zenodo. https://doi.org/10.5281/zenodo.10720027

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
