# Peer review of "GAT TransPruning: progressive channel pruning strategy combining graph attention network and transformer"

_PeerJ Computer Science, doi:10.7717/peerj-cs.2012_

## Round 0.1 · original submission · Major Revisions

Dear authors,


Thank you for submitting your article. The reviewers’ comments are now available. Your article has not been recommended for publication in its current form. However, we encourage you to address the reviewers' concerns and criticisms; particularly regarding readability, experimental design and validity, and resubmit your article once you have updated it accordingly.


When submitting the revised version of your article, it will be better to address the following:

1) The values for the parameters of the algorithms selected for comparison are not given.

2) Explanations of the equations should be checked. All variables should be written in italic as in the equations. Equations should be used with correct equation numbers within the text.

3) Equations are considered part of the text and are punctuated accordingly. If it is the end of a sentence, use a period at the end of the equation; otherwise, use a comma at the end of the equation. Do not capitalize or indent 'where' or 'with' when listing variables following an equation.

4) Some paragraphs are too long to read. They should be divided into two or more for readability and comprehensibility.

5) Clarifying the study’s limitations allows the readers to better understand under which conditions the results should be interpreted. A clear description of limitations of a study also shows that the researcher has a holistic understanding of his/her study. However, the authors fail to demonstrate this in their paper. The authors should clarify the pros and cons of the methods. What are the limitation(s) of the methodology adopted in this work? Please indicate practical advantages, and discuss research limitations.

6) Reviewers have requested that you cite specific references. You may add them if you believe they are especially relevant. However, I do not expect you to include these citations, and if you do not include them, this will not influence my decision.

**Language Note:** The review process has identified that the English language must be improved. PeerJ can provide language editing services - please contact us at [email protected] for pricing (be sure to provide your manuscript number and title). Alternatively, you should make your own arrangements to improve the language quality and provide details in your response letter. – PeerJ Staff

Reviewer 1 ·

Basic reporting

The study, "GAT TransPruning: Progressive channel pruning strategy combining graph attention network and transformer," introduces a novel model compression strategy for neural networks. It combines Graph Attention Network (GAT) and Transformer to achieve effective channel pruning. The study demonstrates promising results in reducing the model size and computational complexity while maintaining accuracy. The methodology includes training an extensive network, pruning, and fine-tuning it to reach a desirable smaller network size. To improve the quality of your study, "GAT TransPruning: Progressive channel pruning strategy combining graph attention network and transformer," the following comments and suggestions are offered:

1. While the introduction outlines the motivation and context effectively, it could benefit from a more precise articulation of the specific challenges and advancements your method addresses in model compression.
2. Consider expanding the literature review to include a broader range of recent advancements in model compression techniques. This would provide a more comprehensive background and highlight the novelty of your approach.
3. Provide a more detailed explanation of the methodology, especially on implementing the GAT and Transformer within the pruning process. This could include step-by-step illustrations or flowcharts for better understanding.
4. Extend the study to include a broader range of network models and datasets. Testing the proposed method on different architectures and more varied datasets will help validate its effectiveness and generalizability.
5. Offer a more in-depth examination of the attention mechanisms used. Explain how these mechanisms function at different network layers and their specific contributions to the pruning process.
6. Provide a more comprehensive analysis of the heatmaps. Discuss how the visualized data correlates with the network's performance post-pruning and how it validates the pruning decisions.
7. Conduct additional ablation studies with various configurations of the GAT and Transformer components. This will help isolate the effects of each component and understand their combined impact better.
8. Integrate more performance metrics such as power efficiency, memory usage, and trade-offs between accuracy and compression. This will give a more holistic view of the model's practicality.
9. Address the computational cost and the algorithmic complexity of your proposed method. This is crucial for understanding its scalability and feasibility in different application scenarios.
10. Elaborate on the practical deployment of the pruned models, particularly in real-world settings like edge computing. Discuss the challenges and potential solutions for deploying these models in resource-constrained environments.
11. Compare your approach with current state-of-the-art methods in both performance and efficiency. This comparison will highlight your method's unique advantages and potential areas for improvement.
12. Pay attention to the overall clarity and quality of writing. Ensure the paper is free from grammatical errors and the technical language is clear and concise. Consider employing professional editing services if necessary.
13. There is a need for significant improvement in the quality of the English language and how it is presented. The text had a notable quantity of typographical errors and grammatical issues. To ensure the quality of your paper, getting assistance from a colleague who is proficient in English and knowledgeable in the subject matter is recommended. Alternatively, you may consider contacting a professional editing service for their expertise.
14. The use of outdated references in the study is valid and essential for maintaining the relevance and currency of the research. I recommend reviewing and incorporating literature from 2021 to 2023 to enhance your study's depth and contemporary relevance. This will provide a more current context for your work and potentially introduce new methodologies, findings, or technologies that could enrich your research. Updating your references to include recent literature is a critical step in strengthening your study's credibility and scholarly value. In light of this, I would like the authors to include this recommended recent literature pertinent to your research and recommend their citation and referencing. In addition, it is possible to have more contemporary literary works published in 2022 and, notably, 2023
a. Qi, M., Cui, S., Chang, X., Xu, Y., Meng, H., Wang, Y.,... Arif, M. (2022). Multi-region Nonuniform Brightness Correction Algorithm Based on L-Channel Gamma Transform. Security and communication networks, 2022. doi: 10.1155/2022/2675950
b. Li, H., Huang, Q., Huang, J., & Susilo, W. (2023). Public-Key Authenticated Encryption With Keyword Search Supporting Constant Trapdoor Generation and Fast Search. IEEE Transactions on Information Forensics and Security, 18, 396-410. doi: 10.1109/TIFS.2022.3224308
c. Liu, H., Xu, Y., & Chen, F. (2023). Sketch2Photo: Synthesizing photo-realistic images from sketches via global contexts. Engineering Applications of Artificial Intelligence, 117, 105608. doi: https://doi.org/10.1016/j.engappai.2022.105608
d. Yang, H., Li, Z., & Qi, Y. (2023). Predicting traffic propagation flow in urban road network with multi-graph convolutional network. Complex & Intelligent Systems. doi: 10.1007/s40747-023-01099-z
e. Dai, W., Zhou, X., Li, D., Zhu, S., & Wang, X. (2022). Hybrid Parallel Stochastic Configuration Networks for Industrial Data Analytics. IEEE Transactions on Industrial Informatics, 18(4), 2331-2341. doi: 10.1109/TII.2021.3096840
f. Liu, X., Lou, S., & Dai, W. (2023). Further results on "System identification of nonlinear state-space models". Automatica, 148, 110760. doi: https://doi.org/10.1016/j.automatica.2022.110760
g. Li, L., & Yao, L. (2023). Fault Tolerant Control of Fuzzy Stochastic Distribution Systems With Packet Dropout and Time Delay. IEEE Transactions on Automation Science and Engineering. doi: 10.1109/TASE.2023.3266065
h. Zhang, R., Li, L., Zhang, Q., Zhang, J., Xu, L., Zhang, B.,... Wang, B. (2023). Differential Feature Awareness Network within Antagonistic Learning for Infrared-Visible Object Detection. IEEE Transactions on Circuits and Systems for Video Technology. doi: 10.1109/TCSVT.2023.3289142
i. Guo, R., Liu, H., & Liu, D. (2023). When Deep Learning-Based Soft Sensors Encounter Reliability Challenges: A Practical Knowledge-Guided Adversarial Attack and Its Defense. IEEE Transactions on Industrial Informatics. doi: 10.1109/TII.2023.3297663
j. Chen, J., Song, Y., Li, D., Lin, X., Zhou, S.,... Xu, W. (2023). Specular Removal of Industrial Metal Objects Without Changing Lighting Configuration. IEEE Transactions on Industrial Informatics. doi: 10.1109/TII.2023.3297613

By addressing these aspects, your study can provide a more comprehensive, validated, and applicable solution in the neural network model compression field, enhancing its academic and practical value.

Experimental design

The study primarily focuses on model compression using VGG and ResNet models with CIFAR-10 and CIFAR-100 datasets, which limits the diversity and may not fully capture the wide range of scenarios applicable to model compression. It lacks a comprehensive comparative analysis with other state-of-the-art model compression techniques, hindering a clear understanding of the method's unique advantages and limitations. There's also an insufficient exploration of different pruning levels and their impact on model performance, accuracy, and efficiency. Real-world application scenarios, particularly in edge computing contexts, are not adequately represented, and the study mainly concentrates on inference speed and accuracy post-pruning, with a need for a broader analysis of performance metrics like energy efficiency and model robustness. The study could benefit from more rigorous statistical analysis, improved reproducibility, and addressing how variability in training data and model initialization affects pruning. Lastly, it lacks an evaluation of pruned models' long-term performance and stability in dynamic conditions or over extended periods.

Validity of the findings

Several factors could potentially weaken the validity of the findings in the study. Some models and datasets, mostly VGG and ResNet models with CIFAR-10 and CIFAR-100 datasets, may not fully capture the wide range of situations that can happen in model compression, which could make the results less useful in real life. Additionally, the absence of a comprehensive comparative baseline with other state-of-the-art model compression techniques limits the ability to contextualize the study's effectiveness within the larger body of research. There is also a lack of in-depth analysis regarding different pruning levels and their specific impacts on model performance, which is crucial for understanding the practical trade-offs of the method. Furthermore, the experimental design overlooks real-world application scenarios, particularly in edge computing, which are critical for assessing the practical applicability of the findings. The focus on a narrow set of performance metrics (primarily inference speed and accuracy) without considering others like energy efficiency or model robustness also limits the comprehensiveness of the evaluation. Lastly, the study could be improved by doing a more thorough statistical analysis, looking into the differences in the training data and how the model was initially set up, and checking how well it worked over time. This would make the conclusions more reliable and useful.

Additional comments

None

Reviewer 2 ·

Basic reporting

No Comment

Experimental design

No Comment

Validity of the findings

No Comment

Additional comments

Annotated reviews are not available for download in order to protect the identity of reviewers who chose to remain anonymous.

---

## Round 0.2 · Minor Revisions

Dear author,

The reviews for your revised manuscript have been received. Your paper still needs minor revision. You will be expected to revise the paper according to the comments on basic reporting provided by Reviewer 1.

Best wishes,

**Language Note:** The review process has identified that the English language must be improved. PeerJ can provide language editing services - please contact us at [email protected] for pricing (be sure to provide your manuscript number and title). Alternatively, you should make your own arrangements to improve the language quality and provide details in your response letter. – PeerJ Staff

Reviewer 1 ·

Basic reporting

After re-reviewing the revised manuscript, I appreciate the authors' efforts in addressing some of the previous concerns. However, there are still several points that require attention:
1. The references used in the study appear to be outdated, with a noticeable lack of articles from the years 2023, 2022, 2021, and so on. The authors must ensure that they include more recent and relevant literature to strengthen the validity and currency of their research findings. Please refer to the previous comment fourteen for further guidance on this matter.
2. The manuscript still contains grammatical and punctuation errors that need to be rectified. To enhance the clarity and professionalism of the article, I recommend seeking the assistance of an English expert for thorough proofreading and editing.
3. It would be beneficial for the authors to upload the code of the implementation to an open repository such as GitHub. This will facilitate research replication and enable other scholars to validate and build upon the findings presented in the study.
Addressing these concerns will significantly improve the quality and credibility of the manuscript. I look forward to seeing the revisions in the next version of the paper.

Experimental design

No comment

Validity of the findings

No comment

Additional comments

No comment

Reviewer 2 ·

Basic reporting

This article have been reviewed previously and all the comment i submitted previously have been addressed.

Experimental design

Comment already addressed

Validity of the findings

Comment already addressed

Additional comments

The Paper is good to go

---

## Round 0.3 · accepted · Accept

Dear authors,

Thank you for the revision and for clearly addressing all the reviewers' comments. I confirm that the paper is improved and addresses the concerns of the reviewers. Your paper is now acceptable for publication in light of this revision.

Best wishes,

Reviewer 1 ·

Basic reporting

After carefully reviewing your manuscript and considering the feedback provided, I'm happy to say that your paper is now good to go for acceptance. You've done a great job of tackling the issues raised and making the recommended changes, which have enhanced the quality of your work. Your efforts have brought your paper up to the standard required for publication. Well, done on your revisions, and thank you for your hard work!

Experimental design

No comment

Validity of the findings

No comment

Additional comments

No comment